# Mix Data or Merge Models?
# Optimizing for Diverse Multi-Task Learning

**Aakanksha**[1], **Arash Ahmadian**[1,2]**, Seraphina Goldfarb-Tarrant**[2],
**Beyza Ermis**[1], **Marzieh Fadaee**[1]**, Sara Hooker**[1]

[1]Cohere For AI [2]Cohere
**Correspondence:** {aakanksha, marzieh, sarahooker}@cohere.com

## Abstract

Large Language Models (LLMs) have been adopted and deployed worldwide for a broad variety of applications. However, ensuring their safe use remains a significant challenge. Preference training and safety measures often overfit to harms prevalent in Western-centric datasets, and safety protocols frequently fail to extend to multilingual settings. We explore model merging in a diverse multi-task setting, combining safety and general-purpose tasks within a multilingual context. Each language introduces unique and varied learning challenges across tasks. We find that objective-based merging is more effective than mixing data, with improvements of up to 8% and 10% in general performance and safety respectively. We also find that language-based merging is highly effective — by merging monolingually fine-tuned models, we achieve a 4% increase in general performance and 7% reduction in harm across all languages on top of the data mixtures method using the same available data. Overall, our comprehensive study of merging approaches provides a useful framework for building strong and safe multilingual models.

## 1 Introduction

Large language models demonstrate strong multitask capabilities, effectively addressing a wide range of tasks across diverse domains [6, 26]. *"Safety"* in a model can be viewed as another "task-solving" ability that a model can learn. It is well established that equipping a model with any kind of capabilities with the standard paradigm of training requires copious amounts of data. Multi-tasking abilities typically arise from fine-tuning models on mixed datasets, which combine data from various sources and across many tasks [28, 40, 50]. However, determining the optimal strategy for mixing datasets in multi-task training is often complex and resource-intensive, as it must ensure that all tasks benefit from the shared training process — especially in the context of safety, where the general performance of models often gets cannibalized in exchange for safety [35, 5, 29, 50].

More recently, an emerging approach for enabling multi-tasking has focused on training distinct models for specific tasks, followed by a weight-merging process governed by a pre-defined algorithm [33, 46, 20, 39, 49, 8]. This method has shown great promise in building models with new capabilities without incurring additional costs and challenges that accompany training from scratch. However, a key question remains – *how does it compare to traditional data mixing and weighting approaches?* In this paper, we explore whether model merging can effectively balance safety and overall performance and how it compares to data mixing techniques, particularly for multilingual alignment.

We evaluate these trade-offs under severe multi-task constraints – optimizing for general and safe performance in a *multilingual setting*. The inherent difficulties of handling multiple languages, each with its unique linguistic structures, cultural nuances, and potential biases, present a formidable task in establishing alignment for these models [30, 19, 17, 37, 18, 50, 3, 32]. Mitigating harm across

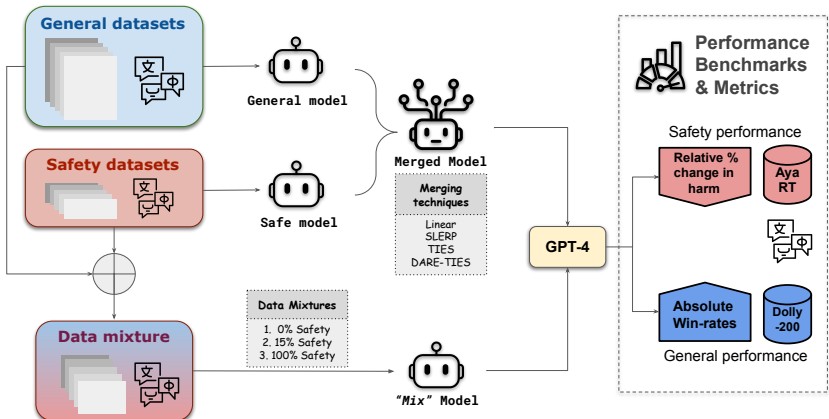

Figure 1: **Overview of our *Mix* versus *Merge* framework:** We analyze the differences in merging models on trained with specialized multilingual datasets, particularly in the context of safety, in contrast to those trained directly on mixtures of these datasets. We follow the LLM-as-a-judge approach for evaluating the performance of these models along two axes – general and safety.

multiple languages is critical given the wide adoption of large models across the world. However, a common issue in safety and alignment work is the narrow focus on addressing safety primarily for English. And so, the challenges are compounded in this scenario by the trivial amount of safety data available across different languages [32]. However, it is precisely because of these severe constraints that this presents an interesting setting to thoroughly evaluate the benefits of merging.

We conduct an exhaustive study to compare traditional approaches for balancing multi-objective training by curating and varying an expansive set of training data mixtures with approaches that merge model checkpoints trained on different subsets of data. Our large-scale evaluation is across six languages from five different language families and encompasses both finetuning and preference training across four different merging techniques. Through our comprehensive experimental setup, we summarize the key findings and contributions of our work as follows:

**1. Merging outperforms mixing**. We find that model merging is more effective than weighting data mixtures for achieving a good balance between safety and generalizability in language models. The top-performing methods for individual objectives were TIES, which reduced harm by 10.4%, and Linear merging, which improved general performance by 8.6% beyond the data mixing approach. The best approach for balancing both objectives was SLERP, which consistently achieved optimal trade-offs across different training strategies, with 3.1% further reductions in harm and 7.0% gains in general performance over the data mixing approach.

**2. Merging is effective at extending multilingual coverage**. We show that merging models across languages is an effective way to manage the dual challenge of safety and multilinguality. Instead of merging across objectives (safety-finetuned model and general-finetuned model), we experiment with merging across languages. Our findings indicate that when each model is trained on a mixture of safety and general data in a single language and then merged, it achieves improvements across both harm reduction and general performance. Specifically, it yields enhancements of up to 3.8% in general benchmarks and a reduction of up to 6.6% in harmful generations compared to a multilingually finetuned model.

**3. Not all merging algorithms are created equal.** We find that not all merging algorithms provide similar performance gains that balance safety and general performance. Some methods consistently result in net positive gains across both axes simultaneously, while others display clear trade-offs between maintaining safe behaviors as well as general-purpose abilities. The highest reduction of harmful generations is achieved by merging DPO checkpoints using the TIES approach, however, this resulted in a decrease of 7.4% in general performance. We see a similar pattern with linear merging as well. Merging models using DARE-TIES and SLERP are more effective at balancing the dual

objectives, with SLERP delivering the most significant improvements in both general performance and harm reduction (7% and 3.1% respectively).

## 2 *Mix* versus *Merge* Setup

In this section, we detail our experimental setup, which involves training models with various data mixtures targeting different objectives to establish the *"Mix"*, followed by merging some of these trained checkpoints into a single model to obtain the *"Merge"*. This setup serves as the foundation for our comprehensive comparison of merging methods' effectiveness in balancing safety and general performance in a multilingual setting. Our experiments cover both supervised fine-tuning (SFT) and offline preference alignment, specifically employing Direct Preference Optimization (DPO) [27].

### 2.1 Merging Approaches

We conduct extensive experiments with diverse data mixtures to create a pool of model candidates. From this pool, we merge the best-performing checkpoints using four different algorithms to produce the final merged models.

**1) Linear Merge:** Linear merging involves simple linear weighted averaging of model parameters, weighted by specified coefficients. This method is widely used in convex optimization and deep learning [24, 38, 43]. This process is formulated as:

$$\theta_{\text{merged}} = \sum_{i=1}^{N} \alpha_i \theta_i \tag{1}$$

where $\alpha_i$ represents the weight assigned to the parameters of each model, with the constraint that $\sum_{i=1}^{N} \alpha_i = 1$. We conduct ablations by varying the values of $\alpha_i$ to investigate different weighting ratios for the base models.

**2) Spherical Linear Interpolation (SLERP):** This technique is used to smoothly blend two models by interpolating their weights along the shortest path on a high-dimensional sphere [42, 11]. SLERP preserves each model's unique characteristics and geometric properties, even in complex spaces. The process involves normalizing the vectors to ensure equal length, calculating the angle $\Omega$ between them, and performing the interpolation as follows:

$$\theta_{\text{SLERP}}(t) = \frac{\sin((1 - t)\Omega)}{\sin(\Omega)} \theta_1 + \frac{\sin(t\Omega)}{\sin(\Omega)} \theta_2 \tag{2}$$

SLERP typically merges only two models at a time. Here, $t \in [0, 1]$ determines the interpolation weight, with $t = 0$ using only *Model 1* and $t = 1$ using only *Model 2*. This method improves upon standard weight averaging by preserving the models' geometric integrity.

**3) TIES-Merging:** This method efficiently combines multiple models by addressing parameter interference and sign conflicts, which occur when models suggest opposing adjustments to the same parameter due to task-specific fine-tuning [45]. The process begins by trimming parameters to retain only those with significant magnitude changes. It then resolves sign conflicts by creating a consensus sign vector:

$$s = \text{sign}\left(\sum_{i=1}^{N} \text{sign}(\theta_i)\right) \tag{3}$$

Finally, it merges the parameters by averaging those that align with the consensus sign:

$$\theta_{\text{merged}} = s \cdot \frac{1}{N} \sum_{i=1}^{N} |\theta_i| \tag{4}$$

TIES-Merging ensures that only parameters contributing to the agreed-upon direction are included in the final model, enhancing performance.

**4) DARE-TIES:** This technique [48] builds upon TIES by applying dropout to the delta parameters before merging them using the TIES method. It reduces interference from redundant parameters and helps maintain the model's overall performance.

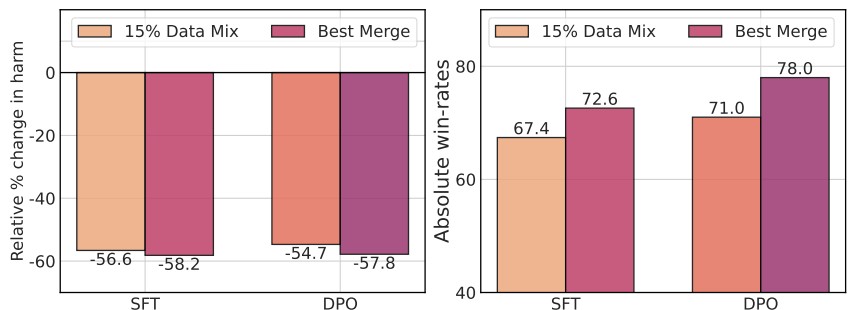

Figure 2: *Mixing versus merging*: Safety and general performance of a *15% Safety Mix* model (§2.2) against SLERP merging, which emerges as the best method for balancing trade-offs, for both SFT and DPO based checkpoints. Lower is better for (a) and higher is better for (b). Both metrics are measured with respect to the Aya 23 base model.

We apply gradient weighting to all merging methods except for Linear Merge. With weighting, we define a blend ratio to specify the merge between the model parameters. Gradient weighting dictates how that ratio changes across the specified values and uses linear interpolation to further establish a smoother gradient of blend ratios for merging the tensors of the models. For example, if the blend ratio between *Model 1* and *Model 2* is defined as [0, 0.5, 1], this implies that the merge begins with 100% of *Model 2's* parameters, gradually transitioning to a 50-50 blend between the two and concluding with only *Model 1's* parameters at the end. For all methods, we conduct an exhaustive search over the set $\{0, 0.3, 0.5, 0.7, 1\}$ to determine the optimal parameter contributions. Our experiments utilize the `mergekit` library from Arcee [11].

## 2.2 Training Data

**Safety dataset.** We use the human-annotated prompts from the multilingual Aya Red-teaming dataset [1] as seeds to synthetically generate pairs of adversarial prompts and contextually safe completions following previous works [1].

**General purpose dataset.** Following previous works [1], we use a sampled set of 10,000 English prompts from the *Ultrafeedback Binarized* [7, 36] dataset translated into our target languages. This dataset will be referred to as the *"general-purpose"* dataset for the remainder of the paper.

**Training data Mix.** We study models trained on different mixtures of data - *0% Safety Mix, 15% Safety Mix* and *100% Safety Mix*. The varying ratio of safety data simulates different objectives – for example, training with 100% safety data allows us to model an upper bound of expected harm mitigation and to obtain a model optimized for safety. In contrast, the 15% Safety mix consists of a combination of safety and general-purpose data in a 1:5 ratio – this represents a more real-world scenario typical of deployment settings. Unless specified otherwise, we use the 15% Safety mix as the baseline for our experimentation. The other mixes follow similar relationships between their naming and ratios.

## 2.3 Key Ablations

In order to study the relative merits of merging for different objectives across a wide set of languages, we conduct extensive ablations. We detail some of the most critical experiment variants below:

**Objective-based merging.** To evaluate the relative merits of merging on balancing dual-objectives, we merge models that have been separately optimized for general-purpose abilities and safety. This builds upon our multilingual 0% and 100% Safety Mixes (see Section 2.2) to balance the trade-offs between safety and general performance.

**Language-based merging.** Multilinguality remains one of the most challenging tasks in language modeling. We aim to determine whether language-specific models can be used off-the-shelf to incorporate language capabilities and explore how merging models based exclusively on different languages affects their downstream performance.

| Type | Method | SFT | | DPO | |
|---|---|---|---|---|---|
| | | Aya RT (↓) | Dolly-200 (↑) | Aya RT (↓) | Dolly-200 (↑) |
| Training data mix | 0% Safety | -41.4 | 70.0 | -39.2 | 70.7 |
| | 15% Safety | -56.6 | 67.4 | -54.69 | 71.0 |
| | 100% Safety | -64.4 | 64.8 | -68.2 | 75.0 |
| Merging | Linear | -49.1 (-7.5) | **76.0** (+8.6) | -48.6 (-6.1) | 75.0 (+4.0) |
| | SLERP | **-58.2** (+1.2) | 72.6 (+5.2) | -57.8 (+3.1) | 78.0 (+7.0) |
| | TIES | -45.2 (-11.4) | 74.9 (+7.5) | **-65.1** (+10.4) | 63.6 (-7.4) |
| | DARE-TIES | -56.1 (-0.5) | 70.0 (+2.6) | -55.9 (+1.2) | **78.5** (+7.5) |

Table 1: Comparison of *Safety* and *General* performance across various methods. *Safety* performance is evaluated using the Aya Red-teaming benchmark [1] in terms of the "Relative Percentage Change in Harmful Generations" while *General* performance is evaluated with the Dolly-200 benchmark as "Absolute Win-rate Percentages". Both metrics are measured with respect to the Aya 23 base model. Scores are aggregated across six languages: English, Hindi, French, Spanish, Arabic, and Russian. Performance deltas, highlighted in color, represent differences from the 15% Safety Mix baseline.

Specifically, we investigate whether combining models optimized for both safety and general performance with a 15% language-specific safety mix for our target languages leads to better performance than training on a mixture of those languages. For clarity, to produce a multilingual model with safe and general-purpose abilities for English, French, and Spanish (referred to as the *EN-FR-SP* group later), we merge models optimized independently on a 15% Safety Mix for each of these languages.

## 2.4 Evaluation

***Baseline***: We evaluate the performance of all models against that of a previous checkpoint of the Aya 23 8B model [3] – which henceforth acts as our baseline for all evaluations. Note that this model was not optimized for safety. We measure the ability to minimize harmful model generations with respect to this model (% decrease). We establish two axes of performance for our experiments — how *safe* model generations are and how well they perform on *general-purpose* benchmarks. We measure these with the following benchmarks:

**1. Safety benchmark**: We use the English prompts from the human-annotated *Aya Red-teaming dataset* [1] and translate them into all of our target languages using the NLLB-3.3B model for an apples-to-apples comparison - i.e., for *Hindi, French, Spanish, Arabic* and *Russian*, resulting in a final set of 6 languages for evaluation. We measure the safety performance on this dataset as the negative relative percent change in harmful model generations with respect to the Aya 23 base model and report aggregated scores over all languages.

**2. General benchmark**: We use the *Multilingual Dolly-200 Eval* set [32, 50], which measures the open-ended generation capabilities of a language model. This dataset consists of a sample of 200 prompts from the Dolly-15k dataset translated into a number of languages, which then acts as a test bed for measuring the general performance of a language model. We use win-rates against the baseline to track performance changes.

To evaluate all experiments, we closely follow the evaluation framework of previous works [1] and use the LLM-as-an-evaluator approach with GPT-4[1] as the judge model. Given our two evaluation axes, safety and general performance, we instruct GPT-4 to classify model outputs as harmful or not to assess safety and to indicate an overall preference between two models' responses (experiment versus the Aya 23 base model) to measure the general performance.

## 3 Results and Discussion

### 3.1 Merging for the win

**Impact on *general* performance.** Merging almost always benefits general performance, with all techniques but one (TIES) outperforming the 15% Safety Mix baseline (see Table 1). We observe

---

[1]https://platform.openai.com/docs/models/gpt-4-turbo-and-gpt-4

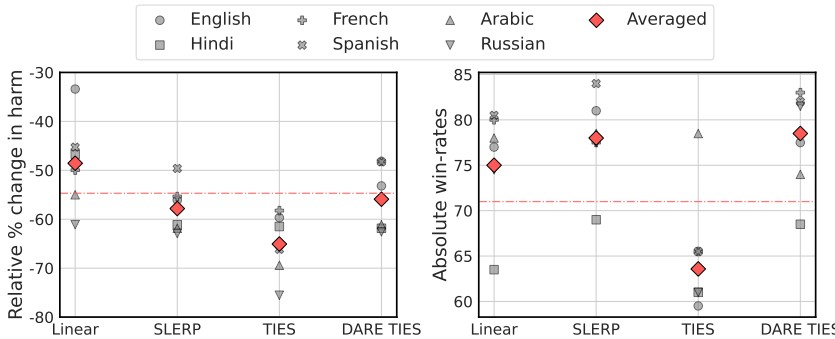

Figure 3: Comparison between different merging methods across safety and general performance with **DPO checkpoints**. Both metrics are measured with respect to the Aya 23 base model. Lower is better for the left and higher is better for the right. The red dashed line represents the model trained on a mixture of safety and general data (*15% Safety Mix*).

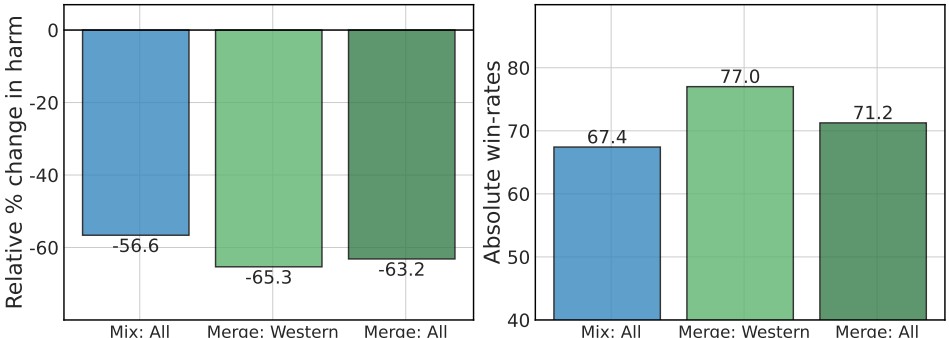

Figure 4: Monolingual model merging: We compare mixing vs merging with SFT checkpoints optimized for languages. The "[All]" bars represent model variants with all 6 languages – *English, Hindi, French, Spanish, Arabic* and *Russian*. "[EN,FR,SP]" represents the pool of *English, French* and *Spanish* "monolingual" models. Both metrics are measured with respect to the Aya 23 base model. Lower is better for the left and higher is better for the right.

gains as high as 7.5% in general performance when combining models with DARE-TIES, closely followed by SLERP with 7% gains.

**Impact on *safety* performance.** Table 1 illustrates that almost all merging methods perform superior to the 15% Safety Mix baseline, with the exception of Linear lagging behind by around 6%, implying that model merging proves beneficial for instilling safety in language models. TIES establishes substantial improvements in harm reduction by around 10% over the 15% Safety Mix.

**Balancing *general* and *safety*.** We evaluate the model with the best trade-off by considering the average percentage change of both objectives relative to the 15% Safety Mix model. Amongst the four methods evaluated, SLERP proved to be the most effective in balancing the two-fold objective of safety and general performance (see Table 1). Figure 2 shows the outcome of SLERP merging for both SFT and DPO checkpoints against the 15% Safety Mix baseline. The model trained on the 15% Safety Mix demonstrates strong performance on general tasks, achieving win rates of 67.4% for SFT and 71% for DPO. However, we see even greater improvements when merging checkpoints, with win-rates rising to 72.6% and 78%, respectively. We observe similar patterns in safety performance — the 15% Safety Mix model reduces harm by 56.6% for SFT and 54.7% for DPO. However, by merging checkpoints instead of mixing data, we achieve further reductions, reaching 58.2% for SFT and 57.8% for DPO. Overall, this supports the claim that merging models explicitly trained for different objectives outperforms building data mixtures aimed at the same goals. This is particularly compelling as a technique given previous studies have shown that optimizing for safety in a language model can negatively impact their general-purpose abilities [5, 29, 4, 50].

## 3.2 DPO merge is better than SFT merge

Given the versatility of merging, which can be applied to any grouping of checkpoints — we compare merging gains when applied to both SFT and DPO (see Table 1). Our experiments show larger consistent improvements when merging DPO checkpoints, with average gains of 2.8% and 2.2% over the base model across the four merging methods assessed for general performance and safety respectively. While merging SFT checkpoints also resulted in significant general performance gains, averaging around 6%, it led to an average increase of 4.6% in harmful generations relative to the 15% Safety Mix model. The best-performing merging approach varies based on the objective (general vs. safety) and underlying training strategy (DPO vs. SFT).

## 3.3 Uneven gains across languages

In this section, we evaluate how merging methods impact different languages. A detailed examination of Figure 3 reveals that although overall improvements are consistent, the optimal trade-offs for different languages depend on the underlying training regime (DPO vs. SFT) of the model checkpoints used for merging.

**Highest beneficiaries.** For DPO, we find that *Russian* shows the most successful safety performance with a reduction of 15% over the 15% Safety Mix model with TIES merging. Spanish exhibits the most impressive improvements with around 6% with SLERP over the 15% Safety Mix baseline in general performance. For SFT, *Hindi* displays the largest reduction in harm (12.14%) with SLERP over the 15% Safety Mix model. However, *Spanish* continues to reap the most benefits from merging with an improvement of 10% gains in general performance with both Linear and TIES.

**Lowest beneficiaries.** Contrary to the above, when merging DPO-based checkpoints, we surprisingly find *English* to benefit the least from merging across both axes of performance. We observe an overall decline of 24.87% in safety and 14.5% in general metrics compared to the 15% Safety Mix model with Linear and TIES merging respectively. For SFT checkpoints in the merging pool, we find that *Spanish* shows the lowest safety performance with TIES with an increase in harmful generations of around 16% while *Hindi* has the least gains in general performance with DARE-TIES with a decline of about 4% in comparison to the 15% Safety Mix.

It is worth noting that while merging leads to performance degradation in some languages compared to data mixing, it still delivers strong results, maintaining an absolute win-rate above 50% for all languages relative to the base model.

## 3.4 Merging monolingual models

Given the challenges posed by multilinguality and the linguistic and cultural variability introduced by each language, especially in the backdrop of safety, we aim to study the impact of merging models exclusively grounded in different languages on their downstream performance. For this set of experiments, we fine-tune our base model, Aya 23 8B, on language-specific data maintaining the 15% Safety Mix (§2.2) and use the resulting checkpoints for merging models across languages. For instance, to obtain a French-only model optimized for both safety and general performance, we fine-tune the model with only French samples, maintaining a 15% mix of safety in the training data. Extending this process for all languages yields 6 separately fine-tuned models on monolingual data.

Additionally, to understand the impact of scaling the number of languages during merging, we combine these models in gradation of two sets: one with 3 languages and another with 6. The 3-language set includes *English, French, Spanish* chosen for their closer familial ties, and is referred to as the *"[EN,FR,SP]"* selection. The 6-language set comprises all our target languages — *English, French, Spanish, Hindi, Arabic* and *Russian* — and is termed *"[All]"* for conciseness henceforth.

We focus on TIES for this set of experiments because its permutation-invariant nature helps us eliminate additional confounders and isolate the impact of language-based merging on overall performance. We use the same baseline as in previous experiments: a fine-tuned version of Aya 23 on a multilingual 15% Safety Mix. Figure 4 presents the results. We find that when compared to the base model, we successfully increase general performance and reduce harm generations across all variants. Merging 6 monolingual models (*"[All]"*) consistently outperforms the corresponding *"mix"* baseline, with safety metrics showing harm reductions as high as 6.6% and absolute improvements of 3.8% in general performance. However, we also observe some evidence of cross-lingual interference;

merging 3 models (*"[EN,FR,SP]"*) yields better performance on both tasks compared to merging 6 models with differences of approximately 2% in safety and 6% in general performance. These results highlight model merging as an effective method for integrating a diverse set of languages without sacrificing performance on key metrics. However, the choice of languages and the number of models significantly influence the performance gains.

## 4 Related Work

**Model Merging.** Recent research has demonstrated success in developing innovative strategies to harness the collective power of multiple LLMs by suggesting methods for combining their unique strengths. This approach offers an efficient solution and has been widely explored for fine-tuned models sharing the same pre-trained base model, thereby sharing a part of their optimization trajectories [10, 15, 14, 43]. Initial efforts focused on merging models with simple weighted averaging of the parameters [43, 23, 12] and showed dramatic performance gains for the resultant merged model. More recently, many works have investigated non-linear methods of merging models [42, 45, 48] while aiming to improve general downstream performance. However, some recent works have focused on ensuring the safety of LLMs when merging, having demonstrated that misalignment transfers trivially from the base to the combined model in this process [13]. Other works "realign" language models by fusing an initial aligned model with many task vectors based on the suitably identified safety subspace [47]. Model merging has also been extended to a multilingual setting – for developing task-solving LLMs for low-resource languages without the availability of SFT data in the target languages [34]. Our work distinguishes itself from prior approaches due to the complexity of the contrasting targets it seeks to satisfy — balancing safety and general-purpose objectives across a wide set of languages. To the best of our knowledge, no prior work has investigated the alignment of LLMs via model merging in a multilingual context while optimizing for a two-fold objective.

**Multilingual Safety.** With the increased pervasiveness of LLMs in recent times, the landscape of language model research has evolved with a heightened emphasis on safeguarding user experiences, thereby placing an increased focus on mitigating potential risks across diverse linguistic contexts. Several works [9, 22] have investigated challenges around multilingual jailbreaks, and introduced novel frameworks and datasets for building robust mitigation strategies. Previous work has examined multilingual toxicity mitigation with a detailed comparison between SFT and retrieval-augmented-based methods [25]. It has been shown that LLMs tend to generate more harmful and irrelevant responses in low-resource languages when prompted maliciously [31]. Techniques such as safety context distillation [50] which harness synthetic data to institute safety guardrails into a model, have shown significant promise towards reducing the harmfulness in model generations. Overall, for a more standardized analysis of safety in multilingual settings, several benchmarks [41, 16, 1] have been introduced and established in recent times. While methods such as SFT and DPO [1, 21] have been studied extensively for aligning language models, some recent works have also pivoted towards weight interpolation for the same objective and have demonstrated the effectiveness of adding a safety vector to compromised fine-tuned models for successful realignment [4]. We direct our efforts towards the development of aligned language models by merging a diverse range of languages.

## 5 Conclusion

In this work, we demonstrated the effectiveness of model merging as a potential solution towards building highly-performant aligned language models across a wide range of languages. Through our comprehensive experimentation, we concluded that models obtained as a result of merging exhibit superior performance on the dual axes of safety and general metrics. However, our experiments also revealed that there is variability in the trade-offs established by different merging algorithms, especially in a multilingual context. Additionally, we also demonstrated the success of combining models to extend language coverage while maintaining performance on the relevant metrics.

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

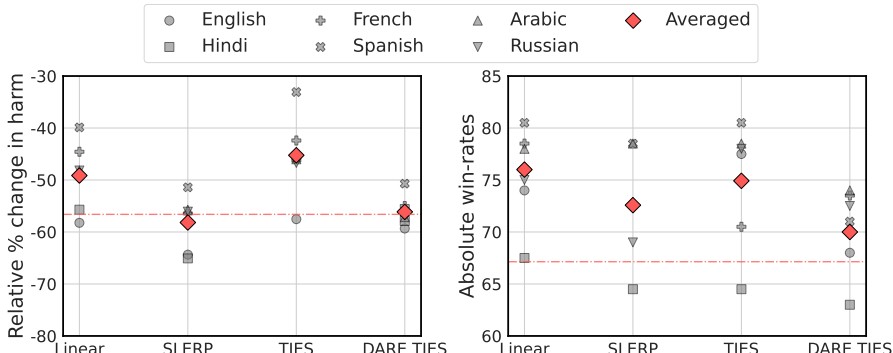

Figure 5: Comparison between different merging methods across safety and general performance with **SFT checkpoints**. Both metrics are measured with respect to the Aya 23 base model. Lower is better for the left and higher is better for the right. The red dashed line represents the model trained on a mixture of safety and general data (*15% Safety Mix*).

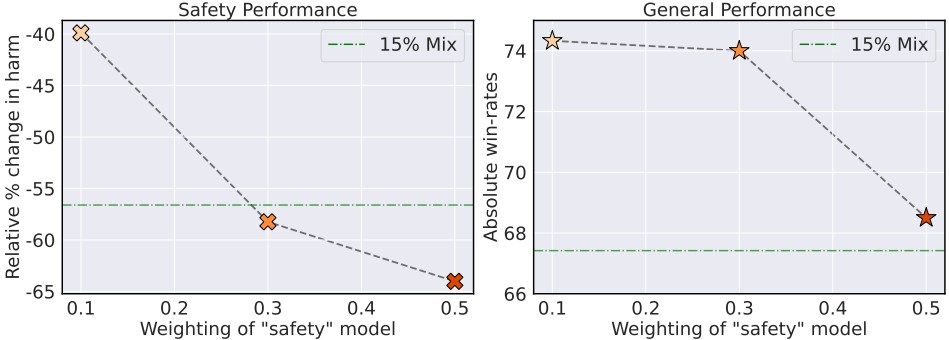

Figure 6: Ablation: Effect of *"safety weighting"* while Linear merging. We vary the weight assigned to the *100% Safety* model while merging linearly and measure the impact of the same. Both metrics are measured with respect to the Aya 23 base model. Lower is better for the left and higher is better for the right.

# A    Additional Ablations

## A.1    Comparison of merging applied to DPO and SFT.

Model merging is a highly adaptable technique that can be applied at any stage of the training process owing to its simple input requirement of model checkpoints. To determine the optimal stage for maximizing its benefits, we merge and evaluate SFT and DPO checkpoints independently as these techniques have shown great success towards the alignment of language models [1, 31].

## A.2    Sensitivity to hyperparameters.

Previous works [14] have shown that merging is sensitive to the hyperparameters involved and have developed sophisticated algorithms [2, 44, 8] to find the optimal values for the same. To this end, we seek to find the impact of varying the weighting scheme of Linear merging on both general performance and safety.

# B    Additional Results

## B.1    DPO merge is better than SFT merge

Given the versatility of merging, which can be applied to any grouping of checkpoints — we compare merging gains when applied to both SFT and DPO (see Table 1). Our experiments show larger

| Training pipeline | Aya RT ($\downarrow$) | Dolly-200 ($\uparrow$) |
|---|---|---|
| SFT → ⟨merge⟩ | -58.2 (+1.6) | 72.6 (+5.2) |
| SFT → DPO → ⟨merge⟩ | -57.8 (+3.1) | **78.0 (+7.0)** |
| SFT → ⟨merge⟩ → DPO | **-61.2 (+6.5)** | 74.0 (+3.0) |

Table 2: Comparison between offline preference tuning models before (row 2) and after (row 3) merging. The scores represent absolute "% relative change in harm" with respect to the Aya 23 base model while the gains in parentheses are reported with respect to the 15% Safety Mix model. The merging technique used here is SLERP.

consistent improvements when merging DPO checkpoints, with average gains of 2.8% and 2.2% over the base model across the four merging methods assessed for general performance and safety respectively. While merging SFT checkpoints also resulted in significant general performance gains, averaging around 6%, it led to an average increase of 4.6% in harmful generations relative to the 15% Safety Mix model. The best-performing merging approach varies based on the objective (general vs. safety) and underlying training strategy (DPO vs. SFT).

## B.2    Impact of safety model weight on merging

Here, we evaluate how model coefficients during merging impact our "objective-based" merging approach on our dual axes of performance. Figure 6 illustrates that the safety performance of the merged model is greatly enhanced when a higher weight is attributed to the safety model. The merged model can mitigate harm more effectively than the 15% Safety Mix baseline, even with a normalized weighting for the constituent safety model as low as 0.3. For general performance, we observe that increasing the weight of the safety-focused model leads to a decrease in the model's performance on general tasks. However, across all weightings, merging models consistently outperforms the data mix run.

## B.3    Continual training after merging

In this section, we examine the dynamics of merging and preference training, focusing on the best ways to integrate both into the training pipeline. More specifically, we use DPO to assess whether continual preference tuning of a merged checkpoint results in stronger models compared to a merged model where the constituent models were individually preference-tuned. As can be seen in Table 2, our experiments demonstrate that continually preference-tuning the models *after* performing the merge yields better outcomes in terms of alignment. The *"after"* merging variant (SFT → ⟨merge⟩ → DPO) shows better safety performance by reducing harmful generations by 6.5% whereas the *"before"* merging variant (SFT → DPO → ⟨merge⟩) exhibits a 3.1% decrease. We observe improvements in the general performance of both variants, with the *"after"* merge variant yielding a 3% increase, and the *"before"* merge variant achieving a 7% increase.

## B.4    Language-based breakdown of "objective-based" merging

Tables 3 - 6 show the language-based breakdown of our "objective-based" merging method.

| Type | Method | English | Hindi | Arabic | French | Spanish | Russian |
|---|---|---|---|---|---|---|---|
| Training data mix | 0% Safety | -58.5 | -46.8 | -41.4 | -33.3 | -32.3 | -34.0 |
| | 15% Safety | -69.1 | -47.3 | -57.2 | -51.4 | -53.5 | -58.1 |
| | 100% Safety | -72.7 | -51.4 | -59.8 | -55.7 | -70.7 | -72.7 |
| Merging | Linear | -58.2 | -55.7 | -48.2 | -44.6 | -39.9 | -48.2 |
| | SLERP | -64.4 | -65.1 | -55.7 | -56.4 | -51.4 | -56.1 |
| | TIES | -57.5 | -45.7 | -46.0 | -42.4 | -33.1 | -46.7 |
| | DARE-TIES | -59.3 | -57.9 | -57.2 | -55.0 | -50.7 | -56.8 |

Table 3: Comparison of *safety* performance with "objective-based merging" across various methods on the Aya Red-teaming benchmark in terms of the "Relative Percentage Change in Harmful Generations" with respect to the Aya 23 base model at a language level. All methods utilize SFT checkpoints.

| Type | Method | English | Hindi | Arabic | French | Spanish | Russian |
|---|---|---|---|---|---|---|---|
| Training data mix | 0% Safety | 68.5 | 57.5 | 76.5 | 73.0 | 77.0 | 67.5 |
| | 15% Safety | 69.5 | 67.0 | 69.0 | 68.5 | 68.5 | 62.0 |
| | 100% Safety | 66.5 | 56.0 | 62.5 | 72.0 | 66.0 | 66.0 |
| Merging | Linear | 74.0 | 67.5 | 78.0 | 78.5 | 80.5 | 75.0 |
| | SLERP | 72.5 | 64.5 | 78.5 | 72.5 | 78.5 | 69.0 |
| | TIES | 77.5 | 64.5 | 78.5 | 70.5 | 80.5 | 78.0 |
| | DARE-TIES | 68.0 | 63.0 | 74.0 | 73.5 | 71.0 | 72.5 |

Table 4: Comparison of *general* performance with "objective-based merging" across various methods on the Multilingual Dolly-200 in terms of "Absolute Win-rates" against the Aya 23 base model at a language level. All values are represent percentages. All methods utilize SFT checkpoints.

| Type | Method | English | Hindi | Arabic | French | Spanish | Russian |
|---|---|---|---|---|---|---|---|
| Training data mix | 0% Safety | -59.1 | -45.6 | -36.5 | -28.7 | -28.6 | -34.4 |
| | 15% Safety | -68.8 | -42.7 | -57.9 | -42.2 | -54.9 | -58.1 |
| | 100% Safety | -76.4 | -62.8 | -61.3 | -62.4 | -67.0 | -77.9 |
| Merging | Linear | -33.4 | -46.7 | -55.0 | -50.0 | -45.3 | -61.1 |
| | SLERP | -56.1 | -61.1 | -61.8 | -55.4 | -49.6 | -62.9 |
| | TIES | -59.7 | -61.5 | -69.4 | -58.2 | -66.2 | -75.5 |
| | DARE-TIES | -53.2 | -61.8 | -61.1 | -48.2 | -48.3 | -62.6 |

Table 5: Comparison of *safety* performance with "objective-based merging" across various methods on the Aya Red-teaming benchmark in terms of the "Relative Percentage Change in Harmful Generations" with respect to the Aya 23 base model at a language level. All methods utilize DPO checkpoints.

| Type | Method | English | Hindi | Arabic | French | Spanish | Russian |
|---|---|---|---|---|---|---|---|
| Training data mix | 0% Safety | 71.5 | 56.0 | 72.0 | 75.0 | 79.5 | 70 |
| | 15% Safety | 74.0 | 61.0 | 71.5 | 73.0 | 78 | 68.5 |
| | 100% Safety | 77.0 | 68.0 | 77.5 | 72.0 | 79.5 | 77 |
| Merging | Linear | 77.0 | 63.5 | 78.0 | 80.0 | 80.5 | 74.5 |
| | SLERP | 81.0 | 69.0 | 79.5 | 77.5 | 84 | 77.5 |
| | TIES | 59.5 | 61.0 | 69.0 | 65.6 | 65.5 | 61.0 |
| | DARE-TIES | 77.5 | 68.5 | 78.5 | 83.0 | 82 | 81.5 |

Table 6: Comparison of *general* performance with "objective-based merging" across various methods on the Multilingual Dolly-200 in terms of "Absolute Win-rates" against the Aya 23 base model at a language level. All values are represent percentages. All methods utilize DPO checkpoints.

