# OpenReview forum: "Mix Data or Merge Models? Optimizing for Performance and Safety in Multilingual Contexts"
_NeurIPS.cc/2024/Workshop/SafeGenAi — SafeGenAi Poster_

### Official Review · Reviewer_sUyp · 2024-10-09
**While the paper provides a good experiment showing the LLMs’ imperfect performance in RPM tests, the paper may need to be revised in some small points to be more rigorous.**

**Rating:** 7
**Confidence:** 4

**Review:**

Thank you for submitting this notable paper. This work experiments by mixing training data and merging various monolingual language models to improve the safety and general performance of multilingual contexts.

It is encouraging to see a relatively comprehensive comparison across different LMs trained in six different languages. The paper also includes extensive experiments with multiple model merging techniques, which shows the effectiveness of model merging over data mixing in achieving a balance between safety and general performance in multilingual settings.

In the abstract, the authors listed several quantitative improvements in safety and general performance but did not expose the specific metrics, that would be better to be added.

It might be better to move the related work section to section 2 of the article since it serves as part of the background information.

---

### Official Review · Reviewer_SMto · 2024-10-12

**Rating:** 5
**Confidence:** 3

**Review:**

The paper investigates the safety of merging pretrained models. The authors show that merging is more preferable than data mixing. In experiments, the authors show improvements in safety and general performance in multilingual inference.

Pros:

(1) The authors provide a nice overview on merging methods.

(2) The authors provide experiments to show safety and general performance of merging and mixing.

Cons:

(1) The authors conduct experimental comparison of existing merging methods. There is no new development of methodology. It would be useful if the authors could elaborate mathematical formulations of merging methods.

(2) It seems that none of method can uniformly perform better than others regarding safety and general performance. It is still not clear how merging methods indeed work for different performance metrics.

(3) It would be more interesting if the authors could include other merging methods and ensembling methods.